

# Measuring blood glucose before or after lumbar puncture

Kaili Shi[1,*], Xue Li[2,*], Ying Li[1], Xiaohua Tan[2], Kelu Zheng[1], Wenxiong Chen[1] and Xiaojing Li[1]

[1] Department of Neurology, Guangzhou Women and Children's Medical Center, Guangzhou, China
[2] Department of Pediatrics, The Third Affiliated Hospital of Guangzhou Medical University, Guangzhou, China
[*] These authors contributed equally to this work.

Corresponding author
Kaili Shi, sklyjh@163.com,
kailihappy@126.com

## ABSTRACT

**Background**. The ratio of cerebrospinal fluid (CSF) to peripheral blood glucose at the same period is an important index for diagnosing and monitoring the efficacy of central nervous system infection, especially bacterial meningitis. Some guidelines refer that blood glucose measurement should be carried out before lumbar puncture. The main reason is to avoid possible effect of stress response induced by lumbar puncture on the level of blood glucose. However, there is no consensus on whether it should be followed in actual clinical work, since up to now no research work having been published on whether lumbar puncture will induce the changes on blood glucose. Our study aimed to investigate the changes of peripheral blood glucose before and after lumbar puncture.

**Methods**. In order to clarify the influence of timing of peripheral blood glucose measurement at the same period of lumbar puncture, a prospective study was conducted including children with an age range from 2 months to 12 years old in the neurology department of a medical center. For those children who need lumbar puncture due to their illness, their blood glucose was measured within 5 min before and after lumbar puncture, respectively. The blood glucose level and the ratio of CSF to blood glucose before and after lumbar puncture were compared. Meanwhile, the patients were divided into different groups according to the factors of sex, age and sedation or not for further comparison. All statistical analyses of the data were performed using SPSS version 26.0 for Windows.

**Results**. In total, 101 children who needed lumbar puncture during hospitalization from January 1, 2021, to October 1, 2021, were recruited with 65 male and 36 female respectively. There was no significant difference on the level of blood glucose, CSF to blood glucose ratio before and after lumbar puncture among the children ($p > 0.05$). No differences were observed within different groups (sex, age, sedation or not) either.

**Conclusion**. It is unnecessary to emphasize blood glucose measurement should be carried out before lumbar puncture, especially for pediatric patients. From the perspective of facilitating smoother cerebrospinal fluid puncture in children, blood glucose measurement after lumbar puncture might be a better choice.

## INTRODUCTION

Acute bacterial meningitis (ABM) is one of the top ten causes of infection-related death worldwide (*Chaudhuri, 2004*). About 30–50% of people who survived after ABM have permanent neurological sequelae (*van de Beek et al., 2004*; *Harnden et al., 2006*). Early use of antibiotics can reduce the incidence and mortality of severe suppurative meningitis. This means early diagnosis is particularly important in clinical work. Cerebrospinal fluid (CSF) detection, such as cell count, glucose, total protein, chloride and culture of CSF, are the most crucial diagnostic basis for suppurative meningitis. The typical abnormalities of CSF in purulent meningitis are mainly the increase of multinucleated leukocytes, the decrease of CSF glucose concentration, the decrease of CSF glucose to blood glucose ratio and the increase of CSF protein level (*van de Beek et al., 2016*). The diagnosis of suppurative meningitis traditionally depends on cerebrospinal fluid culture, although it may take a long time to obtain the bacterial culture. Up to now, Gram staining needs 8–24 h and bacterial culture needs about 72 h. Prolonged diagnosis time is likely to bring adverse consequences to children. In order to make an early diagnosis of suppurative meningitis, perform early treatment, reduce the incidence of sequelae and help the patients to get antibiotics as soon as possible, some researchers suggested to use the CSF/blood glucose ratio, which might be more efficient. It was demonstrated that the CSF/blood glucose ratio may predict the presence of bacterial meningitis more precisely than other routinely measured markers in CSF (*Tamune et al., 2014*). In addition, during the treatment of suppurative meningitis, CSF culture will turn negative. Thus, corresponding indicators are needed to monitor the curative effect. The effect of the treatment can be evaluated by the CSF/blood glucose ratio through continuous lumbar puncture (LP). Therefore, it is essential to obtain blood glucose of the same period of LP appropriately. In 2008, The European Federation of Neurological Societies stated out that blood glucose needs to be obtained before LP in the "EFNS guideline on the management of community-acquired bacterial meningitis: report of an EFNS Task Force on acute bacterial meningitis in older children and adults" (*Chaudhuri et al., 2008*).

Why is it emphasized to obtain blood glucose values before LP? One main reason might be that obtaining blood glucose before LP could exclude the stress response caused by LP that leads to blood glucose increase. Lumbar puncture is a kind of traumatic stress for the human body, which may raise blood glucose. Is it necessary to emphasize obtaining blood glucose before LP? There is no relevant research so far. This guideline is mainly based on theoretical reasoning. That is why there is no consensus for this guideline in many countries. At present, for example, in China some medical institutions follow this guidance strictly, while many others regard it as unnecessary. There is no standardized rule to follow. The purpose of this study was to investigate whether collecting the blood before and after lumbar puncture would cause difference in blood glucose level, and thereby affecting the ratio of CSF glucose/blood glucose. What 's more, we aimed to further clarify whether it is really necessary to measure blood glucose before LP for pediatric patients and thereby providing basic information for actual clinical work.

## MATERIALS & METHODS

This study was conducted in the Department of Neurology, Guangzhou Women's and Children's Hospital from January 1, 2021 to October 1, 2021. A total of 101 children who needed LP during hospitalization were included. This selection was based on the following inclusion criteria: children with no contraindications for LP. Exclusion criteria: patients with endocrine-related diseases; measuring capillary blood glucose for more than 5 min before or after LP; children with sedation failure.

Capillary blood glucose was obtained within 5 min before and after LP by the same person. The blood glucose was measured on the same fingertip of the left ring finger before and after the LP. The same blood glucose meter was used for measurements. The position was sterilized with 75% ethanol and blood glucose measurement was performed after drying. The first drop was discarded and extrusion was minimized. The CSF samples were sent for examination immediately after lumbar puncture in order to avoid degradation of glucose during CSF storage (*Deisenhammer et al., 2006*). The name, inpatient number, sex age, time of capillary blood glucose measurement before and after LP, blood glucose values measured before and after LP, start time of LP, CSF glucose value, state of the patients at the time of puncture (drug sedation, awake) were recorded in detail. All participating researchers have been trained and are qualified.

Data analysis was performed using SPSS statistical software version 26.0. Continuous variables were presented as mean ± SD if normal distribution was established, whereas non-normally distributed measures were expressed as M (P25, P75). To compare changes of blood glucose before and after LP in different age, sex, and conscious state groups, paired *t* test was applied. Wilcoxon signed-ranks test was used for comparing CSF /blood glucose ratio within the groups before and after LP. *P* values less than 0.05 were considered as statistically significant.

This study conforms to the relevant requirements of the Helsinki declaration of the World Medical Association. It was approved by the Ethics Committee of Guangzhou Women's and Children's Hospital (Approval Number: [2022]201A01). Written informed consent for the research was given by the children's guardians in accordance with the recommendations of the biomedical research ethics committees.

## RESULTS

A total of 101 cases, 65 male and 36 female were included in the study, with an age range from 2 months to 12 years . The children were divided into four groups according to their age: group A ($n = 51$): age <3y; group B ($n = 15$): 3y $\leq$ age <6y; group C ($n = 12$): 6y $\leq$ age <9y; group D ($n = 23$): 9y $\leq$ age $\leq$ 12y.

(1) First of all, we compared capillary blood glucose and the CSF/blood glucose ratio of the 101 patients before and after LP (Table 1). There were no significant differences in both

**Table 1 Comparison of blood glucose and the CSF/blood glucose ratio before and after lumbar puncture (LP).**

|  | Before LP | After LP | *z/t* Value | *P* Value |
|---|---|---|---|---|
| Blood glucose mean $\pm$ SD (mmol/L) | 5.89 $\pm$ 0.86 | 5.83 $\pm$ 0.83 | $-1.47$ | 0.15 |
| CSF/blood glucose ratio M (P25, P75) | 0.51 (0.44, 0.60) | 0.52 (0.44, 0.60) | $-1.43$ | 0.15 |

**Notes.**

Normal distributed data are presented by mean $\pm$ SD and non-normally distributed measures are expressed as M (P25, P75).
Paired *t* test and Wilcoxon test were used to analyze the normal distributed and non-normal distributed data, respectively.

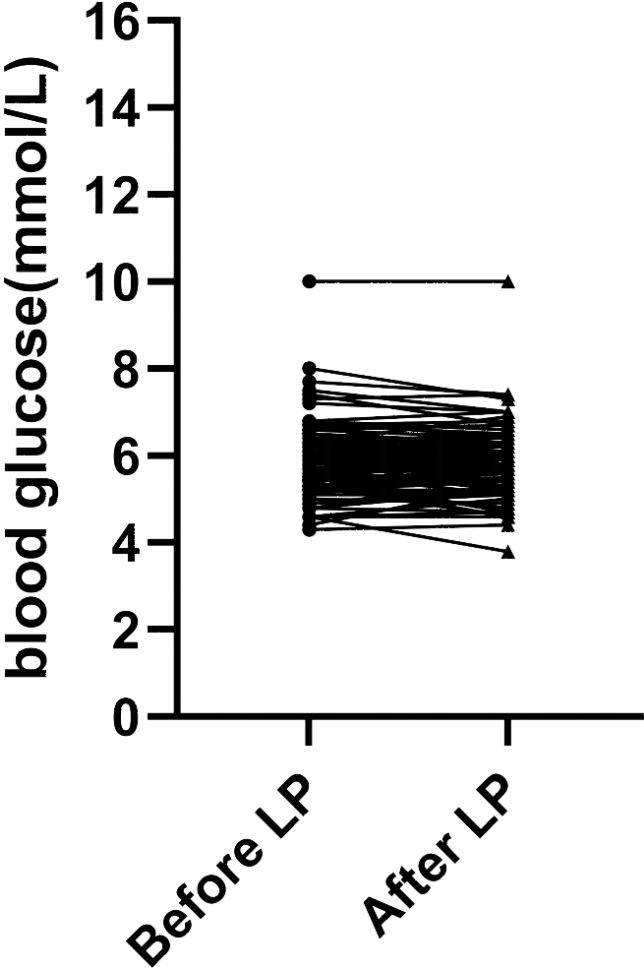

**Figure 1  Blood glucose levels measured before and after lumbar puncture in 101 patients.**

blood glucose levels and the ratio of CSF /blood glucose before and after LP ($p > 0.05$). The trend of blood glucose changes before and after lumbar puncture was shown in Fig. 1.

(2) For different age groups, there were no significant differences of blood glucose and the CSF/blood glucose ratio before and after LP ($p > 0.05$) (Table 2).

(3) No significant differences in the blood glucose level and the CSF/blood glucose ratio before and after LP were observed for different sexes ($p > 0.05$) (Table 3).

**Table 2 Comparison of blood glucose and the CSF/blood glucose ratio before and after lumbar puncture (LP) in different age groups of children.**

| Group | Number | Blood glucose | | | | CSF/Blood glucose ratio | | | |
|---|---|---|---|---|---|---|---|---|---|
| | | Before LP mean ± SD (mmol/L) | After LP mean ± SD (mmol/L) | t Value | P Value | Before LP M (P25, P75) | After LP M (P25, P75) | z Value | P Value |
| A (age<3y) | 51 | 5.81 ± 0.63 | 5.80 ± 0.61 | −0.13 | 0.90 | 0.49 (0.42, 0.54) | 0.48 (0.43, 0.56) | −0.26 | 0.80 |
| B (3y ≤age<6y) | 15 | 5.71 ± 0.90 | 5.61 ± 0.98 | −0.97 | 0.35 | 0.60 (0.53, 0.65) | 0.60 (0.52, 0.70) | −0.56 | 0.57 |
| C (6y ≤age<9y) | 12 | 6.10 ± 1.09 | 5.92 ± 0.86 | −1.88 | 0.09 | 0.60 (0.49, 0.65) | 0.60 (0.50, 0.67) | −1.49 | 0.14 |
| D (9y ≤age ≤12y) | 23 | 6.08 ± 1.12 | 6.01 ± 1.10 | −1.08 | 0.29 | 0.52 (0.40, 0.59) | 0.49 (0.38, 0.60) | −1.19 | 0.24 |

Notes.
Normal distributed data are presented by mean ± SD and non-normally distributed measures are expressed as M (P25, P75). Paired t test and Wilcoxon test were used to analyze the normal distributed and non-normal distributed data, respectively.

**Table 3 Comparison of blood glucose and the CSF/blood glucose ratio before and after lumbar puncture (LP) in different sex groups of children.**

| Group | Number | Blood glucose | | | | CSF/blood glucose ratio | | | |
|---|---|---|---|---|---|---|---|---|---|
| | | Before LP mean ± SD (mmol/L) | After LP mean ± SD (mmol /L) | t Value | P Value | Before LP M (P25, P75) | After LP M (P25, P75) | z Value | P Value |
| Males | 65 | 5.88 ± 0.82 | 5.82 ± 0.83 | −1.28 | 0.21 | 0.50 (0.45, 0.58) | 0.51 (0.46, 0.59) | −1.58 | 0.11 |
| Females | 36 | 5.91 ± 0.94 | 5.86 ± 0.83 | −0.74 | 0.46 | 0.56 (0.40, 0.62) | 0.56 (0.41, 0.64) | −0.48 | 0.63 |

Notes.
Normal distributed data are presented by mean ± SD and non-normally distributed measures are expressed as M (P25, P75). Paired t test and Wilcoxon test were used to analyze the normal distributed and non-normal distributed data, respectively.

**Table 4 Comparison of blood glucose and the CSF/blood glucose ratio before and after lumbar puncture (LP) in sedated and awake children.**

| Group | Number | Blood glucose | | | | CSF/blood glucose ratio | | | |
|---|---|---|---|---|---|---|---|---|---|
| | | Before LP mean ± SD (mmol/L) | After LP mean ± SD (mmol /L) | t Value | P Value | Before LP M (P25, P75) | After LP M (P25, P75) | z Value | P Value |
| Drug sedation | 59 | 5.93 ± 0.77 | 5.87 ± 0.73 | −1.26 | 0.21 | 0.50 (0.43, 0.60) | 0.52 (0.43, 0.60) | −1.35 | 0.18 |
| Awake | 42 | 5.84 ± 0.98 | 5.78 ± 0.96 | −0.83 | 0.41 | 0.51 (0.45, 0.58) | 0.52 (0.45, 0.61) | −0.94 | 0.35 |

Notes.
Normal distributed data are presented by mean ± SD and non-normally distributed measures are expressed as M (P25, P75). Paired t test and Wilcoxon test were used to analyze the normal distributed and non-normal distributed data, respectively.

(4) The patients were divided into two groups according to their state of consciousness at the time of lumbar puncture, drug sedation and awake group. No significant changes in blood glucose and the CSF/blood glucose ratio were found before and after LP between the groups ($p > 0.05$) (Table 4).

## DISCUSSION

Analysis of CSF is a crucial tool for the diagnosis of central nervous system infections, especially suppurative meningitis. CSF glucose is one of the most important indicators for CSF analysis. It plays an irreplaceable role in the diagnosis and identification of infectious diseases of the nervous system. In bacterial meningitis, the bacteria multiply and consume the glucose of the CSF. In addition, the impaired function of the glucose transport system also results in a decrease in CSF glucose (*Pimentel & Hansen, 2001*; *Nigrovic et al., 2012*). However, CSF glucose mainly comes from blood through the blood–brain barrier. Plasma glucose has a significant impact on CSF glucose (*Nigrovic et al., 2012*; *Verbeek et al., 2016*). The CSF glucose levels are directly proportional to the plasma levels. It means that we should not assess CSF glucose levels only, simultaneous measurement of blood glucose is required (*Deisenhammer et al., 2006*). Studies have shown that the CSF/blood glucose ratio is an effective indicator for the diagnosis of bacterial meningitis, thereby could be used as a marker and speed up the physician's decision to administer antibiotics (*Tamune et al., 2014*; *Briem, 1983*; *Lindquist et al., 1988*; *Rousseau et al., 2017*). The CSF/ blood glucose ratio could be used for differential diagnosis and efficacy evaluation as well. Obtaining the correct ratio of CSF to blood glucose is critical.

The article 'EFNS guideline on the management of community-acquired bacterial meningitis: report of an EFNS Task Force on acute bacterial meningitis in older children and adults' by the European Federation of Neurological Societies in 2008 declared that blood glucose should be obtained before LP (*Chaudhuri et al., 2008*). The stress response caused by LP may affect blood glucose. It is well known that acute injury can produce stress response to body and lead to increasing blood glucose (*Marik & Bellomo, 2013*). When the body subjected to traumatic stimulation, it will induce series of complex microcirculation, neuroendocrine and other pathophysiological changes. Regulation of neuroendocrine system manifested as sympathetic-adrenal medulla and the hypothalamic-pituitary-adrenal cortex system caused excessive gluconeogenesis, glycogenolysis and insulin resistance, resulting in elevated blood glucose (*Marik & Bellomo, 2013*; *Barth et al., 2007*; *Korakas et al., 2022*; *Bae & Ahn, 2022*). As a stimulus, will LP cause stress response and increase blood glucose? Should we measure blood glucose before LP as suggested in the guideline (*Chaudhuri et al., 2008*)?

Based on our results, there were no significant differences in capillary blood glucose levels and the ratio of CSF/blood glucose before and after LP for children between 2 months to 12 years old. Considering that age, sex and state of consciousness may have different response to stress, the tested patients were grouped and compared. None of these factors have affected the measured results. The stress stimulation of LP was not able to cause changes in capillary blood glucose. Blood glucose measurement after LP did not alter the blood glucose level and the ratio of CSF /blood glucose. Regarding to the applicability, for pediatric patients, LP is usually performed under drug sedation. Measuring blood glucose before LP may wake up the sedated children and will increase the difficulty of LP, hence increase the failure rate of LP. From the perspective of facilitating smoother cerebrospinal fluid puncture in children, blood glucose measurement after lumbar puncture might be

a better choice. Additionally, our results could also provide useful information in the diagnosis of other neurological and genetic pathologies, such as the glucose transporter type 1 deficiency syndrome.

## CONCLUSIONS

It is unnecessary to emphasize the obtention of peripheral blood glucose of pediatric patient before lumbar puncture, especially for the patient who are sedated before LP. Taken together, our study provides a basis for the formulation of relevant guidelines.

## ACKNOWLEDGEMENTS

We would like to thank the participants and their parents of this trial. We thank all colleagues who have contributed to this research.

### Funding
The authors received no funding for this work.

### Competing Interests
The authors declare there are no competing interests.

### Author Contributions
- Kaili Shi conceived and designed the experiments, performed the experiments, analyzed the data, authored or reviewed drafts of the article, and approved the final draft.
- Xue Li conceived and designed the experiments, performed the experiments, analyzed the data, prepared figures and/or tables, and approved the final draft.
- Ying Li conceived and designed the experiments, prepared figures and/or tables, and approved the final draft.
- Xiaohua Tan performed the experiments, prepared figures and/or tables, and approved the final draft.
- Kelu Zheng performed the experiments, authored or reviewed drafts of the article, and approved the final draft.
- Wenxiong Chen performed the experiments, authored or reviewed drafts of the article, and approved the final draft.
- Xiaojing Li conceived and designed the experiments, authored or reviewed drafts of the article, and approved the final draft.

### Data Availability
   The raw data are available in the Supplementary File.

### Supplemental Information
Supplemental information for this article can be found online at http://dx.doi.org/10.7717/peerj.15544#supplemental-information.

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
