# Peer review of "Measuring blood glucose before or after lumbar puncture"

_PeerJ, doi:10.7717/peerj.15544_

## Round 0.1 · original submission · Minor Revisions

As you will see, all reviewers identified your work as worthy of publication, but have raised some points for clarification, and requested that your data be deposited in the public domain. Please address all comments in your revision and outlined in your cover letter how these have been addressed in the text.

·

Basic reporting

English proofreading is required.

Experimental design

no comment

Validity of the findings

no comment

Additional comments

Thank you for offering me the opportunity to review this exceptional research paper on the necessity of blood glucose measurement prior to lumbar puncture for the calculation of the CSF/blood glucose ratio. The research question and its answer have been presented with utmost clarity and simplicity. Overall, I believe that this manuscript deserves publication in PeerJ. However, I would like to comment on certain aspects that could be improved before publication.

#1. I recommend the inclusion of line plots that depict glucose changes before and after lumbar puncture, as shown in the following reference: https://community.rstudio.com/t/line-graph-before-and-after-treatment/11980.
The coding language can be in the authors' preferred one. Such information would provide a comprehensive overview of the raw data. If the authors prefer, the line plots can include many extra data such as Men/Women difference, age preference, and so on.

#2. There are several abbreviations used in the Tables, including SD, M, Q1, Q3, LP, z/t Valve, and P Valve (P value?). I suggest that the authors spell out all abbreviations to avoid any ambiguity.

#3. In Table 2, please add a column indicating the age range of the study population.

#4. Additionally, I recommend that the authors have a colleague who is proficient in English and familiar with the subject matter review the manuscript to correct any typos, grammar errors, or sentences that may be unclear. Alternatively, the authors may consider contacting a professional editing service.

·

Basic reporting

The article is written clearly, with accurate terms and no ambiguity in the description. The literature review and references are broad enough and suitable for the research purpose. The overall structure is good, and the tables are clear. All the results necessary for verifying the hypothesis are described in the manuscript.

Experimental design

The experimental design was appropriate, according to the ethical parameters to be considered for this type of research. The research was approved by the ethics committee and has the necessary informed consent.
It is suggested to describe what type of analysis was used for the study of the age categories (a total of 4): how were the groups compared? and was ANOVA used as part of the applied tests or not? This is considered taking into account that the tests described (T and Wilcoxon) only allow for 2 variables simultaneously (lines 107-109).

Validity of the findings

The findings are important in clarifying a frequent procedural doubt regarding lumbar puncture. The results are clear, and conclusive according to the analysis performed, and allow for easily reaching conclusions, which are clearly described in the corresponding section. The impact of the results goes beyond the diagnosis of neuroinfection and could be useful in the diagnosis of other neurological and genetic pathologies. Therefore, it is suggested to mention this in the discussion.

Additional comments

According to the methods described, the analysis was carried out from capillary glucose samples (glucometry), which may have high variability due to factors such as blood flow, temperature, among others, making it more difficult to evaluate the cerebrospinal fluid glucose / blood glucose relationship. This should be taken into account for the discussion. For greater clarity, it is suggested to refer to capillary glucose in the writing, instead of blood glucose.

Reviewer 3 ·

Basic reporting

In the manuscript, Shi et al, have reported that blood glucose levels do not change after lumbar puncture in patients, which does not affect the CSF glucose to blood glucose levels, which is widely used as an indication for detecting bacterial infections. The manuscript is well reported with raw data also provided.

Experimental design

The design is well reported, with blood glucose levels before and after the lumbar puncture, as well different parameters (sedation, age group etc)

Validity of the findings

The findings are interesting, and limited to guidelines that have been previously observed. While the arguments of why this finding is pertinent and advantageous (blood glucose level monitored after LP) are not discussed in detail, the data supports the conclusion.

Since the manuscript aims to report a single finding of no change in blood glucose levels, I have no further comments.

---

## Round 0.2 · accepted · Accept

Thank you for careful consideration of the reviewers suggestions. I am happy to recommend acceptance.